# Role of Age, Comorbidity, and Frailty in the Prediction of Postoperative Complications After Surgery for Vulvar Cancer: A Retrospective Cohort Study with the Development of a Nomogram

**DOI:** 10.3390/curroncol32010021

**Published:** 2024-12-31

**Authors:** Giovanni Delli Carpini, Francesco Sopracordevole, Camilla Cicoli, Marco Bernardi, Lucia Giuliani, Mariasole Fichera, Nicolò Clemente, Anna Del Fabro, Jacopo Di Giuseppe, Luca Giannella, Enrico Busato, Andrea Ciavattini

**Affiliations:** 1Gynecologic Section, Department of Odontostomatologic and Specialized Clinical Sciences, Università Politecnica delle Marche, 60131 Ancona, Italy; giovanni.dellicarpini@ospedaliriuniti.marche.it (G.D.C.); c.cicoli@pm.univpm.it (C.C.); marco.bernardi@ospedaliriuniti.marche.it (M.B.); l.giuliani@pm.univpm.it (L.G.); m.fichera@pm.univpm.it (M.F.); jacopo.digiuseppe@ospedaliriuniti.marche.it (J.D.G.); luca.giannella@ospedaliriuniti.marche.it (L.G.); 2Gynecological Oncology Unit, IRCCS CRO, Centro di Riferimento Oncologico, National Cancer Institute, 33081 Aviano, Italy; drfrancesco@sopracordevole.it (F.S.); nicolo.clemente@cro.it (N.C.); anna.delfabro@cro.it (A.D.F.); 3Department of Gynaecology and Obstetrics, Treviso Regional Hospital, AULSS 2 Marca Trevigiana, Piazzale Ospedale 1, 31100 Treviso, Italy; enrico.busato@aulss2.veneto.it

**Keywords:** vulvar neoplasms, surgery, postoperative complications, comorbidity, frailty, prognosis

## Abstract

Surgery is the cornerstone of vulvar cancer treatment, but it is associated with a significant risk of complications that may impact prognosis, particularly in older patients with multiple comorbidities. The objective of this study was to evaluate the role of age, comorbidities, and frailty in predicting postoperative complications after vulvar cancer surgery and to develop a predictive nomogram. A retrospective cohort study was conducted, including patients who underwent surgery for vulvar cancer at two Italian institutions from January 2018 to December 2023. A logistic regression model for the rate of Clavien-Dindo 2+ 30-days complications was run, considering the age-adjusted Charlson Comorbidity Index (AACCI), body mass index (BMI), and frailty as exposures. Lesion characteristics and surgical procedures were considered as confounders. Among the 225 included patients, 50 (22.2%) had a grade 2+ complication. The predictive score of the nomogram ranged from 44 to 140. The AACCI (0–64 points) and BMI (0–100 points) were independently associated with a risk of complications. A nomogram including the AACCI and BMI predicts the risk of complications for patients undergoing surgery for vulvar cancer. The preoperative determination of the risk of complications enables surgical planning and allows a tailored peri- and postoperative management plan.

## 1. Introduction

Vulvar cancer, though infrequent, predominantly affects older patients, with incidence rates rising with age [1,2,3]. This age-related increase is further complicated by multiple comorbidities and frailty, often affecting treatment choices and postoperative outcomes [4,5,6]. Indeed, frailty encompasses physical vulnerability and reduced physiological reserve, which may heighten the susceptibility to surgical stress and adverse events [7,8]. Surgery is the cornerstone of vulvar cancer treatment [7]. Despite advances in surgical techniques that have sought to reduce invasiveness, postoperative complications remain common and often severe. These complications may have profound implications, decreasing patient quality of life and delaying the initiation of adjuvant therapies, potentially impacting survival outcomes [2,9,10,11,12]. Therefore, preoperative assessment is essential for surgical outcome optimization by assessing the patient-specific overall surgical risk. A practical, easy-to-use clinical tool to predict specific complication risks remains unavailable [1,6]. Even if traditional frailty and comorbidity scores provide valuable insights, they often lack a consideration of the body mass index (BMI), which is a growing concern, given its established role in increasing surgical risk, particularly in gynecologic oncology [5,7,8,13]. In addition to patient characteristics, disease-specific variables are essential for assessing complication risks [5,14]. The development of a simple tool combining age, comorbidities, frailty, BMI, and tumor characteristics would represent a significant advancement in individualized patient care. The objective of our study is, therefore, to evaluate the roles of age, comorbidities, frailty, BMI, and tumor characteristics in predicting postoperative complications after vulvar cancer surgery. Additionally, we aim to develop a predictive nomogram incorporating patient and disease factors to assist in preoperative risk stratification and support more tailored surgical planning for this vulnerable population.

## 2. Materials and Methods

This was a retrospective cohort study including patients affected by vulvar cancer managed at two Italian institutions: 1. Gynecologic Section, Department of Odontostomatologic and Specialized Clinical Sciences, Università Politecnica delle Marche, Ancona, Italy, and 2. Gynecological Oncology Unit, IRCCS CRO—Centro di Riferimento Oncologico—National Cancer Institute, Aviano, Italy from January 2018 to December 2023. All patients with a histopathological diagnosis of vulvar cancer, eligible for primary surgery, were included. Patients were excluded in case of locally advanced or advanced vulvar cancer not suitable for a surgical approach, contraindications to surgery due to comorbidity or frailty, previous diagnosis of vulvar cancer, or primary treatment by radiotherapy or chemo-radiotherapy.

According to local institutional protocols and international guidelines [7], all patients with a histopathological diagnosis of vulvar cancer made by punch or incision biopsy referred to our institutions were evaluated with the following approach: 1. medical history; 2. general assessment with evaluation and categorization of comorbidities; 3. frailty assessment; 4. clinical examination with palpation of the inguinofemoral lymph nodes; 5. vulvar examination to evaluate tumor focality (unifocal/multifocal), size, site, mobility, distance to the midline (medial border < 1 cm or >1 cm from midline), and distance/infiltration of urethra/vagina/anus of the dominant lesion; to perform additional separate biopsies in case of multiple lesions; and to take comprehensive photographs of the vulvar region; and 6. imaging (ultrasound examination of inguinofemoral lymph nodes, chest-abdomen computed tomography, and magnetic resonance imaging in case of locally advanced tumors). The choice of treatment was made in a multidisciplinary setting. Patients suitable for surgical approach underwent radical local excision of all primary lesions, unilateral or bilateral lymphadenectomy (superficial and deep femoral nodes with preservation of the saphenous vein) for tumors with >1 mm invasion at the preoperative biopsy according to the distance of the medial margin to the midline, and reconstructive surgery with V-Y flap according to the characteristics of the patient and the lesion. After hospital discharge, a clinical examination was performed 30 days after surgery or in case of onset of symptoms that were suspected of postoperative complications. The pathology report included specimen and tumor dimensions, the histopathological type, the depth of invasion, the tumor margin status, the presence/absence of lymph vascular space invasion, the presence/absence of perineural invasion, the lymph node status, and the total number of involved nodes [15]. The pathological staging was defined according to the International Federation of Gynecology and Obstetrics (FIGO) classification [16]. The multidisciplinary team decided on postoperative management (re-excision, follow-up, or adjuvant treatment by radiotherapy or chemo-radiotherapy).

The following variables were collected from clinical charts: age (in years, both as continuous variable and by classes: <50 years, 50–59 years, 60–69 years, 70–79 years, and ≥80 years), BMI (in kg/m^2^, both as continuous variable and by classes: BMI ≤ 30 and BMI > 30), age-adjusted Charlson comorbidity index (AACCI) (both as continuous variable and by classes: AACCI < 4 and AACCI ≥ 4), 5-factor modified frailty index (mFI-5) (classes: not frail mFI-5 0–1 and frail mFI-5 ≥ 2), localization of dominant vulvar lesion (medial: medial margin < 1 cm from the midline; and lateralized: medial margin > 1 cm from the midline), surgical procedures performed (vulvar excision, unilateral or bilateral inguinofemoral lymphadenectomy, and reconstructive surgery), postoperative overall complications at 30 days according to the Clavien–Dindo classification [17], vulvar complications, inguinofemoral complications, definitive histopathology (diameter of dominant lesion in mm, surgical margins status, lymph node status, and number of lymph nodes), FIGO stage, and adjuvant treatment (radiotherapy/chemo-radiotherapy). To calculate the AACCI, we first assigned 1–6 points to 5 age classes (0 point: <50 years; 1 point: 50–59 years; 2 points: 60–69 years; 3 points: 70–79 years; and 4 points ≥ 80 years) and 19 comorbidities, if present (1 point: myocardial infarction, congestive heart failure, peripheral vascular disease, cerebrovascular disease, dementia, chronic obstructive pulmonary disease, connective tissue disease, peptic ulcer disease, mild liver disease, and uncomplicated diabetes; 2 points: hemiplegia, moderate or severe renal disease, diabetes with organ damage, solid tumor without metastases, leukemia, and lymphoma; 3 points: moderate or severe liver disease; and 6 points: metastatic solid tumor and acquired immune deficiency syndrome). The final AACCI score was determined by summing all the assigned points [18,19]. The mFI-5 was determined by assigning 1 point to each of the following five conditions, if present: diabetes mellitus, hypertension requiring medication, chronic obstructive pulmonary disease, congestive heart failure, and functional dependency [5].

The primary outcome was the rate of postoperative complications at 30 days, categorized as dichotomic: no complication/Clavien–Dindo grade 1 and Clavien–Dindo grade 2+. BMI, AACCI, and mFI-5 were considered as exposures. Age, dominant lesion localization (medial/lateralized), dominant lesion size (mm), execution of bilateral lymphadenectomy, execution of reconstructive surgery, FIGO stage III/IV, positive tumor margins, positive lymph nodes, and the number of lymph nodes were considered as confounders.

### Statistical Analysis

The G*Power version 3.1.9 software was used to determine the required sample size using the method described by Hsieh et al. for logistic regression with a binomial outcome (primary outcome: Clavien–Dindo 0–1/Clavien–Dindo 2+) [20]. According to previous literature, the rate of Clavien–Dindo 2+ postoperative complications after surgery for vulvar cancer ranges between 25.5% and 31.3% [2,5,10]. For sample size determination, we considered a mean value of 27.5% (H0 = 0.275). We assumed that the rate of postoperative Clavien–Dindo 2+ complications should double under the effect of the considered exposures (Age, BMI, AACCI, and mFI-5) to be considered significantly different from the general population of patients affected by vulvar cancer (H1 = 0.55). Considering α = 0.05, a power of 0.95, an R2 other than X of 0.25 for the confounders (moderate association), and an X parm Π of 0.5, the resulting total sample size was 222 patients. We included patients from January 2018 to December 2023 to reach the required sample size. Statistical software R (version 4.4.1—R Foundation for Statistical Computing, Vienna, Austria) was used for data analysis. All continuous variables were tested for normality with the Shapiro–Wilk test. Normally distributed variables were expressed as mean ± standard deviation (SD), while not-normally distributed variables were reported as median and interquartile range (IQR). Qualitative variables were expressed as numbers and proportions. All the collected variables were compared between exposed (BMI > 30, AACCI ≥ 4, and frail) and unexposed (BMI ≤ 30, AACCI < 4, and not frail) in a bivariate analysis. The t-test, the Mann–Whitney test, or the Chi-square test were used for comparison as appropriate. A *p* < 0.05 was considered statistically significant. None of the considered variables had the characteristics of an instrumental variable [21]. A logistic regression with a binomial dependent variable was run from the primary outcome (Clavien–Dindo grade 2+ postoperative complications). We included the following variables as predictors: BMI, AACCI, Frail (mFI-5 ≥ 2), age, lateralized dominant lesion, dominant lesion size, bilateral lymphadenectomy, reconstructive surgery, FIGO stage III/IV, positive tumor margins, positive lymph nodes, and number of lymph nodes. An adjusted OR with 95% CI for the primary outcome was determined for each included predictor. Predictors that were significantly and independently associated with the primary outcome with a *p* < 0.05 were selected for nomogram construction. Variables were represented in the nomogram as line segments with varying lengths according to weight, with scores ranging from 0 to 100. The total score predicted the risk of Clavien–Dindo 2+ complications.

## 3. Results

During the study period, 225 patients subjected to primary surgery for vulvar cancer in the two institutions were included in the analysis. The median (IQR) age was 75 (66–82) years, with 16 (7.1%) patients < 50 years, 21 (9.3%) 50–59 years, 44 (19.1%) 60–69 years, 74 (32.9%) 70–79 years, and 72 (31.6%) > 80 years. The mean ± SD BMI was 26.1 ± 4.7, and the median (IQR) AACCI was 4 (3–5). The median (IQR) mFI-5 was 1 (0–1), with 52 (23.1%) patients categorized as frail. The dominant vulvar lesion was defined as lateralized in 47 (20.9%) cases. The surgical approach consisted of a radical local excision of all primary lesions for all included patients. Inguinofemoral lymphadenectomy was performed in 197 (87.6%) patients. Among those, 154 (78.2%) patients underwent a bilateral lymphadenectomy, and 43 (21.8%) had a unilateral lymphadenectomy. Reconstructive surgery was performed in 17 (7.5%) cases. The histopathological stage, according to the FIGO staging, was IA for 8 (3.6%) patients, IB for 138 (61.3%), II for 14 (6.2%), IIIA for 41 (18.2%), IIIB for 17 (7.6%), IIIC for 4 (1.8%), and IVB for 3 (1.3%). According to the Clavien–Dindo classification, no surgical complication at 30 days was reported in 169 (75.2%) patients, 1 (0.4%) patient had a grade I complication, 47 (20.9%) a grade II complication, 3 (1.3%) a grade IIIA complication, and 5 (2.2%) a grade IVA complication. Vulvar complications were reported in 29 (12.9%) cases, and inguinofemoral complications in 20 (10.2%) of the 197 patients subjected to inguinofemoral lymphadenectomy. The median (IQR) maximum diameter of the lesions at histopathology was 30 (18–40) mm. A positive margin was found in 38 (16.9%) cases. The median (IQR) number of removed lymph nodes was 14 (11–18). At least one positive lymph node was found in 60/197 (30.5%) cases. A total of 143 (63.6%) patients were sent to follow-up, 32 (14.2%) underwent a second surgical approach to obtain negative margins, and 51 (22.6%) were subjected to radiotherapy, 4 (1.8%) to chemotherapy, and 21 (9.3%) to chemo-radiotherapy.

Table 1, Table 2 and Table 3 report the comparison between exposed and unexposed subjects. Binomial logistic regression was performed to ascertain the effects of the BMI, AACCI, mFI-5, age, dominant lesion localization, dominant lesion size, execution of bilateral lymphadenectomy, execution of reconstructive surgery, and FIGO stage III/IV on the risk of postoperative Clavien–Dindo 2+ complications (primary outcome). The logistic regression model was statistically significant, χ^2^ (12) = 86.2, *p* < 0.001. The model explained 33.0% (McFadden R^2^) of the variance in the rate of Clavien–Dindo 2+ postoperative complications. Of the nine included predictor variables, only two were independently associated with the risk of postoperative complications: the BMI and AACCI (Table 4).

The nomogram’s predictive score ranged between 45 and 140, corresponding to a risk of postoperative complications of 5–95%. The weights ranged from 0 to 100 for BMI values between 14 and 42 (*p* < 0.001) and from 0 to 64 for AACCI values between 0 and 10 (*p* < 0.001). The final version of the nomogram is reported in Figure 1.

## 4. Discussion

The results of this study showed that age, comorbidities (evaluated by the AACCI), and BMI are independent predictors of postoperative complications after surgery for vulvar cancer. The preoperative evaluation of those factors may help predict the risk of complications and adapt clinical management using a simple clinical tool.

Those findings align with and expand upon the existing literature on the topic. Several studies have documented the impact of age and associated comorbidity on surgical outcomes in older patients, underscoring the complexity of managing vulvar cancer in this population [1,11,13,19]. Our study builds on these findings by highlighting the additive predictive value of the body mass index (BMI), a factor often underrepresented in traditional comorbidity assessments. The relationship between the BMI and surgical morbidity is increasingly recognized, particularly in studies focusing on inguinofemoral lymphadenectomy for vulvar cancer. Cacciamani et al. (2023) and Gitas et al. (2021) both reported that higher BMI levels are linked to increased rates of wound complications, infections, and lymphedema in patients undergoing lymph node dissection [8,11]. Additionally, Wills and Obermair (2013) found that obesity contributes to higher complication rates, a finding echoed by Rahm et al. (2022) and Sotelo et al. (2024), who noted the BMI as a predictor of adverse outcomes across similar surgical contexts [2,6,13].

The predictors related to the lesion or the surgical procedures were not independently associated with the risk of complications, showing that those factors may have a less significant role than age, comorbidities, or frailty. This lack of association could be due to the fact that the patients included in the study underwent a standardized surgery; thus, these surgical factors exerted their effect equally among all included patients. Indeed, the same standardized approach for all included patients (the excision of vulvar lesions rather that extended vulvectomy, choice for unilateral lymphadenectomy when indicated, removal of inguinofemoral lymph nodes according to vulvar embryology and anatomy, and preservation of the saphenous vein [22,23]) may have determined a more significant role of additional factors in determining the risk of complications.

The lack of association (*p* = 0.083) at the logistic regression between frailty assessed by mFI-5 and the main outcome may be related to the significant overlap of the conditions assessed by mFI-5 with those assessed by the AACCI (4/5 parameters).

An important observation from our study is that the combined evaluation of comorbidities and age (AACCI) increase the likelihood of adverse outcomes. This finding aligns with the population-based study by Wills and Obermair (2013), which demonstrated that complication rates escalate with patient age and comorbidity burden, particularly in complex gynecologic surgeries like those required for vulvar cancer [13]. The expert consensus by Sotelo et al. (2024) emphasizes using standardized metrics for evaluating peri-operative complications, particularly in high-risk surgeries [6].

Prediction tools in the form of nomograms are available for vulvar cancer in terms of prognosis estimation. However, the nomogram proposed in this study is the first to specifically focus on predicting 30-day postoperative complications [24]. The development of a predictive nomogram for postoperative complications that incorporates age, comorbidities, and BMI represents a promising step toward personalized surgical management for vulvar cancer patients. The inclusion of the nomogram in a multidisciplinary setting for vulvar cancer management is of crucial importance, with the possibility of including this instrument in “case-by-case” discussions [25]. A tool of this nature could help clinicians stratify patients based on their postoperative risk profile, allowing for adjustments in the extent of surgical intervention and applying preventive strategies tailored to individual needs [11,26]. For example, patients identified as high-risk might benefit from less invasive surgical techniques or more intensive postoperative monitoring to mitigate complication risks [27]. A significant advancement in reducing surgical morbidity is the sentinel lymph node (SLN) biopsy, which spares patients with early stage, unifocal tumors from complete lymphadenectomy. Studies by Rahm et al. (2022) and Gitas et al. (2021) emphasize the efficacy of the SLN biopsy in reducing complications compared to more extensive lymph node dissection [2,11]. However, the SLN biopsy is not universally applicable, particularly for patients with larger or multifocal tumors requiring complete lymphadenectomy [8]. An additional surgical procedure that should be considered in the surgical planning is the V-Y flap reconstruction, which is associated with a reduced rate of complications according to Di Donato et al., particularly in vulnerable patients [10].

While a targeted surgery is needed, greater attention should be directed towards optimizing the patient’s general condition before surgery. Indeed, patients affected by vulvar cancer may benefit from preoperative interventions, such as nutritional support or physical conditioning [28].

This study is strengthened by the sample size and the rigorous methodology focused on outcome prediction according to exposures and confounders. Even if two different institutions were involved, the external validation of our nomogram in different clinical settings may be advisable. Future studies should evaluate the outcome of vulvar cancer patients according to this nomogram and identify additional predictive factors. Moreover, prospective studies should investigate preoperative measures to reduce surgical complication. The retrospective nature of our study may be a limitation, considering the risk of selection bias associated with potential unmeasured confounders, like socioeconomic status. Regarding selection bias, its effect may have been limited, considering that the characteristics of our study population in terms of the role of exposures are similar to the characteristics reported in other studies about vulvar cancer patients [10,11,14,19] or patients affected by different types of gynecological cancers [29].

## 5. Conclusions

The preoperative evaluation of the AACCI and BMI is crucial in predicting postoperative complications. Those parameters can be easily inserted in an easy-to-use tool to assist in preoperative risk assessment, enhancing the capacity of personalized surgical care in this population. Identifying high-risk patients before surgery allows us to tailor the oncological approach to reduce complications and improve patient outcomes. The nomogram could also be converted into a straightforward web-based interface for instant risk calculation simply by inputting the parameters of the AACCI and BMI of any identified patient.

## Figures and Tables

**Figure 1 curroncol-32-00021-f001:**
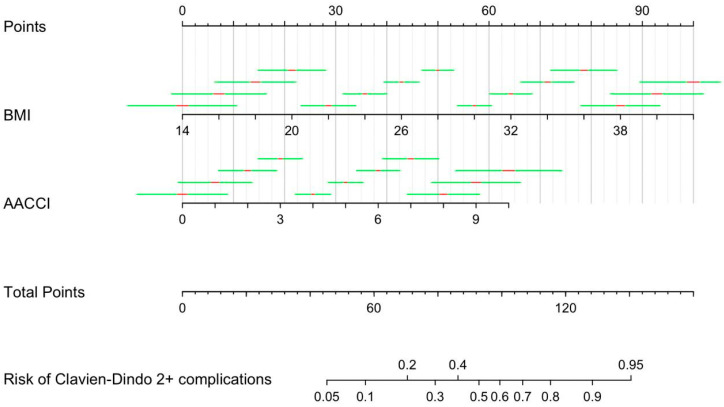
Nomogram for prediction of Clavien–Dindo 2+ postoperative complications.

**Table 1 curroncol-32-00021-t001:** Bivariate analysis according to the exposure BMI.

Variable	BMI ≤ 30 (n = 179)	BMI > 30 (n = 46)	*p*
Age	75 (66–82)	74 (66–81)	0.712
AACCI	3 (2–5)	4 (4–5)	0.016
AACCI ≥ 4	107 (59.8)	35 (76.1)	0.041
Frail (mFI-5 ≥ 2)	34 (19.0)	18 (39.1)	0.004
Lateralized dominant lesion	35 (19.6)	12 (26.1)	0.331
Bilateral lymphadenectomy *	129/172 (75.0)	25/25 (100)	0.005
Reconstructive surgery	8 (4.5)	9 (19.6)	<0.001
Diameter of dominant lesion (mm)	27 (18–35)	35 (21–40)	0.029
Positive lymph node *	53/172 (30.8)	7/25 (28.0)	0.775
Stage III–IV	45 (25.1)	20 (43.5)	0.014
**Outcome**			
Clavien–Dindo 2+	35 (19.6)	20 (43.5)	<0.001
Vulvar complications	15 (8.4)	14 (30.4)	<0.001
Positive margins	27 (15.1)	11 (23.9)	0.154
Inguinofemoral complications *	15/172 (8.7)	5/25 (20.0)	0.081
Number of lymph nodes *	14 (10–18)	14 (13–17)	0.386

* Among the 172 patients in whom lymphadenectomy was performed.

**Table 2 curroncol-32-00021-t002:** Bivariate analysis according to the exposure AACC.

Variable	AACCI < 4 (n = 83)	AACCI ≥ 4 (n = 142)	*p*
Age	74 (66–81)	75 (66–82)	0.515
BMI	24.1 ± 4.8	27.2 ± 4.3	<0.001
BMI > 30	11 (13.3)	35 (24.6)	0.041
Frail (mFI-5 ≥ 2)	6 (7.2)	46 (32.4)	<0.001
Lateralized dominant lesion	16 (19.3)	31 (21.8)	0.649
Bilateral lymphadenectomy *	56/77 (72.7)	98/120 (81.7)	0.138
Reconstructive surgery	3 (3.6)	14 (9.9)	0.087
Diameter of dominant lesion (mm)	30 (17–40)	30 (20–40)	0.649
Positive lymph node *	23 (29.9)	37 (30.8)	0.886
Stage III–IV	15 (18.1)	50 (35.2)	0.006
**Outcome**			
Clavien–Dindo 2+	11 (13.3)	44 (31.0)	0.003
Vulvar complications	8 (9.6)	21 (14.8)	0.266
Positive margins	12 (14.5)	26 (18.3)	0.457
Inguinofemoral complications *	2/77 (2.6)	18/120 (15.0)	0.005
Number of lymph nodes *	14 (11–18)	14 (11–17)	0.468

* Among the 172 patients in whom lymphadenectomy was performed.

**Table 3 curroncol-32-00021-t003:** Bivariate analysis according to the exposure mFI-5.

Variable	Not Frail (mFI-5 0–1) (n = 173)	Frail (mFI-5 ≥ 2) (n = 52)	*p*
Age	73 (64–82)	78 (72–85)	0.002
BMI	25.8 ± 4.5	27.1 ± 5.4	0.067
BMI > 30	28 (16.2)	18 (34.6)	0.004
AACCI	3 (2–4)	5 (4–6)	<0.001
AACCI ≥ 4	96 (55.5)	46 (88.5)	<0.001
Lateralized dominant lesion	42 (24.3)	5 (9.6)	0.023
Bilateral lymphadenectomy *	127/154 (82.5)	27/43 (62.8)	0.006
Reconstructive surgery	11 (6.4)	6 (11.5)	0.215
Diameter of dominant lesion (mm)	28 (18–38)	30 (19–50)	0.358
Positive lymph node *	47/154 (30.5)	13/43 (30.2)	0.971
Stage III–IV	48/173 (27.7)	17/52 (32.7)	0.490
**Outcome**			
Clavien–Dindo 2+	32 (18.5)	23 (44.2)	<0.001
Vulvar complications	20 (11.6)	9 (17.3)	0.278
Positive margins	26 (15.0)	12 (23.1)	0.174
Inguinofemoral complications *	11/154 (7.1)	9/43 (20.9)	0.008
Number of lymph nodes *	14 (11–18)	13 (10–15)	0.024

* Among the 172 patients in whom lymphadenectomy was performed.

**Table 4 curroncol-32-00021-t004:** Binomial logistic regression for Clavien–Dindo 2+ postoperative complications.

Predictor	Estimate	SE	Z	*p*	OR	95% CI
BMI	0.39325	0.0778	5.055	<0.001	1.482	1.27–1.73
AACCI	0.48091	0.1546	3.111	0.002	1.618	1.20–2.19
Frail (mFI-5 ≥ 2)	0.93358	0.5377	1.736	0.083	2.544	0.89–7.30
Age	0.01173	0.0190	0.619	0.536	1.012	0.98–1.05
Lateralized dominant lesion	−0.73131	0.5986	−1.222	0.222	0.481	0.15–1.56
Dominant lesion size	1.15 × 10^−5^	0.0140	0.000	0.999	1.000	0.97–1.03
Bilateral lymphadenectomy	0.35139	0.7161	0.491	0.624	1.421	0.35–5.78
Reconstructive surgery	0.63451	0.6911	0.918	0.359	1.886	0.49–7.31
FIGO stage III/IV	−0.76572	0.5084	−1.506	0.132	0.465	0.17–1.26
Positive tumor margins	−1.04996	0.6398	−1.641	0.101	0.350	0.10–1.23
Positive lymph nodes	−0.78559	0.5220	−1.505	0.132	0.456	0.16–1.27
Number of lymph nodes	−0.00617	0.0365	−0.169	0.866	0.994	0.92–1.07
Intercept	−13.96783	2.7605	−5.060	<0.001	0.00	0.00–0.00

## Data Availability

The raw data supporting the conclusions of this article will be made available by the authors upon request.

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
