# Peer review of "Role of Age, Comorbidity, and Frailty in the Prediction of Postoperative Complications After Surgery for Vulvar Cancer: A Retrospective Cohort Study with the Development of a Nomogram"

_curroncol, 2024, doi:10.3390/curroncol32010021_

Round 1
Reviewer 1 Report
Comments and Suggestions for Authors
This study offers a robust review of predictors of postoperative complications in vulvar cancer surgery, addressing gaps in current literature.
Focusing on age, comorbidities, frailty, and BMI, the authors develop a practical nomogram that significantly enhances preoperative risk stratification and personalized surgical care for a vulnerable population.
It is also well-structured with clear objectives, methodology, and data come from two institutions, thus strengthening generalizability. The nomogram is user-friendly and aligns with previous studies which outline the prognostic value of frailty in surgical outcomes.
Although the study is of value,
- Better explanation of the external validation of the nomogram would make it even more useful.
- The study relies on retrospective data, thus limiting the ability to prove a direct causal relationships between the predictors (AACCI, BMI, frailty) and postoperative complications and therefore the introduction of selection bias.
- Certain confounding factors, such as socio-economic conditions, were not measured in the model but could impact outcomes. The insignificance of surgical and lesion characteristics as predictors is interesting. Do you think there is an explanation?
- One could mention in the discussion also the importance of a multidisciplinary approach, which is directly relevant to how a nomogram could be integrated into clinical workflows. Tagliaferri et al. (2020) (https://doi.org/10.1136/ijgc-2020-001465) share the same goal of the authors which is personalized medicine. Referring to the article could strengthen the argument that predictive tools are valuable in tailoring patient care within multidisciplinary settings: the multidisciplinary context reinforces the idea that nomograms are not standalone tools but part of a broader decision-making framework.
Overall, this work covers an important topic in gynecologic oncology, and it is well-suited for publication in Current Oncology with some few corrections.
Author Response
Reviewer 1
Comments and Suggestions for Authors
This study offers a robust review of predictors of postoperative complications in vulvar cancer surgery, addressing gaps in current literature. Focusing on age, comorbidities, frailty, and BMI, the authors develop a practical nomogram that significantly enhances preoperative risk stratification and personalized surgical care for a vulnerable population. It is also well-structured with clear objectives, methodology, and data come from two institutions, thus strengthening generalizability. The nomogram is user-friendly and aligns with previous studies which outline the prognostic value of frailty in surgical outcomes.
Thank you for the appreciation of our work!
Although the study is of value,
- Better explanation of the external validation of the nomogram would make it even more useful.
Thank you for this comment. The process of external validation will be a crucial step in improving the generalizability of our results. We hope that future studies will be in this direction. We expanded the discussion section on this aspect (Page 8, line 270-274).
- The study relies on retrospective data, thus limiting the ability to prove a direct causal relationships between the predictors (AACCI, BMI, frailty) and postoperative complications and therefore the introduction of selection bias.
Thank you for this comment. Indeed, the retrospective nature of the study implicate a potential selection bias. However, the rate of exposed and unexposed of our study is similar to that reported in other studies on the topic, thus limiting the impact of selection bias, since our sample is similar to the general population of patients affected by vulvar cancer or gynecological cancers. More specifically, the rate of frail patients in our study is 23.1%, similar to the range of 25% reported by Levine et al. in 2023 and of 24.1% reported by Reiser et al. in 2020 for patients affected by different type of gynecological cancers; the mean ± SD BMI in our study was 25.8 ± 4.5, similar to the values reported by Di Donato et al. in 2019 (BMI 27.5 ± 5.5), by Di Donato et al. in 2024 (BMI 27.4 ± 5.3 and BMI 26.6 ± 5.7), by Gans et al. in 2023 (BMI 28.1 ± 6.6), by Gitas et al. in 2021 (BMI 27.5 ± 6.1), and by Reiser et al. in 2020 (BMI 24.0 ± 4.3); our median AACCI of 4 (3-5) was slightly greater than the mean AACCI values reported by Di Donato et al. in 2024 (2.98 and 3.18). We added the discussion of this potential limitation in the manuscript (Discussion section, Page 8, line 275-279).
- Certain confounding factors, such as socio-economic conditions, were not measured in the model but could impact outcomes. The insignificance of surgical and lesion characteristics as predictors is interesting. Do you think there is an explanation?
Unfortunately, data about socio-economic conditions were not available in our dataset. Therefore, we added the example of “socioeconomic status” as a potential unmeasured confounder in the Discussion section (Page 8, line 275-276). The lack of significant association between surgery or lesion characteristics on the main outcome is indeed interesting. We believe that this may be due to the standardized approach to surgery adopted in the two institutions, with the aim of limiting complications, such as the tailored excision of vulvar lesions (rather than a wider approach like partial or total vulvectomy), the choice for unilateral lymphadenectomy when indicated, the choice for an anatomy-based lymphadenectomy with removal of lymph nodes located between the boundaries of superficial inguinofemoral lymphadenectomy (laterally: circumflex vein; medially: external pudendal vein) and of the lymph nodes located medially to the femoral and saphenous veins, within the first 3 cm of the saphenous vein, and the preservation of the saphenous vein. A standardized approach applied in all included patients may have determined a more significant role of additional factors in determining the risk of complications. We expanded the comment about this aspect in the Manuscript (Page 7, line 226-232).
- One could mention in the discussion also the importance of a multidisciplinary approach, which is directly relevant to how a nomogram could be integrated into clinical workflows. Tagliaferri et al. (2020) (https://doi.org/10.1136/ijgc-2020-001465) share the same goal of the authors which is personalized medicine. Referring to the article could strengthen the argument that predictive tools are valuable in tailoring patient care within multidisciplinary settings: the multidisciplinary context reinforces the idea that nomograms are not standalone tools but part of a broader decision-making framework.
Thank you for this insightful suggestion. We added a comment about this aspect in the Discussion section (Page 7, line 248-250).
Overall, this work covers an important topic in gynecologic oncology, and it is well-suited for publication in Current Oncology with some few corrections.
Reviewer 2 Report
Comments and Suggestions for Authors
In this interesting study the authors developed a nomogram to predict 30 days postop morbidity in vulvar cancer patients all treated by a radical local excision and inguinofemoral nodal dissection (in 90%). The objective was to develop this nomogram and to propose/discuss tailored surgical treatment depending on the results of this nomogram.
There are several issues I would like to address:
1. Why do the authors use 2 co-morbidity scales (AACCI and MFI-5) that show significant overlap in co-morbidities, such as "diabetes" and "hypertension". Why not using one method?
2. In the results section (page 4, line 191) the authors state that 3/9 (BMI;AACCI;mFI-5) variables were independantly associated with postop complications. However, when analysing Table 4, I notice that the 95% CI for mFI-5 (>=2) is 0,89-7.30 with a p-value of 0.083. That means this is not an independant predictor. Please correct this, or otherwise explain why this is mentioned "an independant predictor".
3. The authors should better explain the clinical consequences of using the nomogram in order to tailor treatment. They mention using the SLN as a method to decrease postop morbidity (page 7, lines 249-258). But this method is already used as a standard method for patients with unifocal tumor with a diameter less than 4 cm and clinically non- suspicious inguinal nodes. So, to be honoust, we do not need this nomogram for that patient category. Therefore the important question is: what are the clinical (surgical) consequences for patients at high risk for postop complications (according to the nomogram) who are not suitable for a SLN procedure? Most of the time these are patients with more advanced disease, therefore the question is what kind of modifications (tailoring) in surgical therapy do you recommend for that specific group of patients?
Author Response
Reviewer 2
Comments and Suggestions for Authors
In this interesting study the authors developed a nomogram to predict 30 days postop morbidity in vulvar cancer patients all treated by a radical local excision and inguinofemoral nodal dissection (in 90%). The objective was to develop this nomogram and to propose/discuss tailored surgical treatment depending on the results of this nomogram.
There are several issues I would like to address:
- Why do the authors use 2 co-morbidity scales (AACCI and MFI-5) that show significant overlap in co-morbidities, such as "diabetes" and "hypertension". Why not using one method?
Thank you for this comment. Indeed, the two scales have a significant overlap, which may explain the lack of association of mFI-5 (see comment below).
- In the results section (page 4, line 191) the authors state that 3/9 (BMI;AACCI;mFI-5) variables were independantly associated with postop complications. However, when analysing Table 4, I notice that the 95% CI for mFI-5 (>=2) is 0,89-7.30 with a p-value of 0.083. That means this is not an independant predictor. Please correct this, or otherwise explain why this is mentioned "an independant predictor".
Thank you for this insightful observation. In the first draft of the manuscript, we chose to include all factor with a p < 0.1 rather than < 0.05, in order to not excessively limit the exploration of the predictive factors of complications. However, including only factors with a p < 0.05 may be more representative of the true contribution of AACCI and mFI-5, also considering the overlap between the two scales. We modified the manuscript and Figure 1 accordingly, only mentioning the role of AACCI and BMI (see also Page 7, line 233-235).
- The authors should better explain the clinical consequences of using the nomogram in order to tailor treatment. They mention using the SLN as a method to decrease postop morbidity (page 7, lines 249-258). But this method is already used as a standard method for patients with unifocal tumor with a diameter less than 4 cm and clinically non- suspicious inguinal nodes. So, to be honoust, we do not need this nomogram for that patient category. Therefore the important question is: what are the clinical (surgical) consequences for patients at high risk for postop complications (according to the nomogram) who are not suitable for a SLN procedure? Most of the time these are patients with more advanced disease, therefore the question is what kind of modifications (tailoring) in surgical therapy do you recommend for that specific group of patients?
Thank you for this comment. We believe that a standardized and tailored surgical approach is the first crucial step in the management of vulvar cancer patients. This is confirmed from the fact that surgical characteristics were not associated with post-operative complications at the logistic regression, thus the effect of comorbidity scales was more evident. This may be due to the fact that the factors related to the surgical technique had the same effect on all included patients, confirming the necessity of a rigorous and standardized approach. The standardized approach should include a tailored excision of vulvar lesions (rather than a wider approach like partial or total vulvectomy when not needed), the choice for an unilateral lymphadenectomy when indicated, an anatomy-based lymphadenectomy (removal of superficial lymph nodes located between the lateral boundary -circumflex vein- and the medial boundary -external pudendal vein- and of the lymph nodes located medially to the femoral and saphenous vein, within the first 3 cm of the saphenous vein), and the preservation of the saphenous vein. Therefore, we recommend that strict adherence to these surgical principles is fundamental, as well as pre-operative intervention on comorbidities (when feasible) and more intensive surveillance after surgery for high-risk patients. Moreover, a more extensive use of V-Y flap reconstruction may be advisable, since it is reported in literature that is associated with lower risk of complications. Additionally, it should not overlooked that not all institutions may have access to the SLN equipment, therefore an upgrade of the technology or the referral to different institutions may also be advised. We expanded the discussion about these topics in the Discussion section (Page 7, line 226-232; Page 8, line 261-263).
Round 2
Reviewer 2 Report
Comments and Suggestions for Authors
The authors sufficiently answered the questions. No further comments.